# Targeting the Viral Polymerase of Diarrhea-Causing Viruses as a Strategy to Develop a Single Broad-Spectrum Antiviral Therapy

**DOI:** 10.3390/v11020173

**Published:** 2019-02-20

**Authors:** Marcella Bassetto, Jana Van Dycke, Johan Neyts, Andrea Brancale, Joana Rocha-Pereira

**Affiliations:** 1School of Pharmacy and Pharmaceutical Sciences, Cardiff University, CF10 3NB Cardiff, UK; bassettoM@cardiff.ac.uk (M.B.); brancaleA@cardiff.ac.uk (A.B.); 2KU Leuven—Laboratory of Virology and Chemotherapy, Department of Microbiology and Immunology, Rega Institute for Medical Research, University of Leuven, Leuven 3000, Belgium; jana.vandycke@kuleuven.be (J.V.D.); johan.neyts@kuleuven.be (J.N.)

**Keywords:** RNA viruses, DNA viruses, viral polymerase, antiviral, norovirus, rotavirus

## Abstract

Viral gastroenteritis is an important cause of morbidity and mortality worldwide, being particularly severe for children under the age of five. The most common viral agents of gastroenteritis are noroviruses, rotaviruses, sapoviruses, astroviruses and adenoviruses, however, no specific antiviral treatment exists today against any of these pathogens. We here discuss the feasibility of developing a broad-spectrum antiviral treatment against these diarrhea-causing viruses. This review focuses on the viral polymerase as an antiviral target, as this is the most conserved viral protein among the diverse viral families to which these viruses belong to. We describe the functional and structural similarities of the different viral polymerases, the antiviral effect of reported polymerase inhibitors and highlight common features that might be exploited in an attempt of designing such pan-polymerase inhibitor.

## 1. Introduction: The Need and the Concept of Antiviral Treatment for Viral Diarrhea

Acute gastroenteritis (AGE) occurs worldwide and is an important health issue that affects all age groups. Human rotaviruses (HRV), dsRNA viruses belonging to the *Reoviridae* family, are the most important cause of viral AGE in children <5 years, annually resulting in 215,000 deaths [1]. After the introduction of two rotavirus vaccines, human noroviruses (HNoV), (+)ssRNA viruses belonging to the *Caliciviridae* family, have been gaining impact in this age group, becoming in fact the most common viral agent of AGE in all age groups, resulting in 200,000 deaths per year [2]. Other human diarrhea-causing viruses such as astroviruses (HAstV, *Astroviridae*, (+)ssRNA), sapoviruses (HSaV, *Caliciviridae*, (+)ssRNA) and adenoviruses type 40 and 41 (HAdV, *Adenoviridae*, dsDNA) are common agents of AGE in young children, causing more moderate disease and being less prevalent than rota- and noroviruses. Infection with these viruses occurs via the fecal-oral route, by ingestion of contaminated food or water and by person-to-person contact. Large outbreaks occur frequently, disturbing the functioning of health institutions, long-term care facilities and other semi-closed environments [3].

AGE manifests suddenly, with vomiting and watery non-bloody diarrhea being the most common symptoms. Differentiating the causative agent based on symptoms alone is unfeasible, thus specific diagnostic tests are required. Although AGE is normally self-limiting, it can be debilitating, with prolonged and severe disease occurring in vulnerable populations, namely young children, the elderly and immunocompromised individuals. There are currently no specific antiviral drugs available and treatment is limited to supportive therapy with oral rehydration salts (ORS). Outbreak control and prevention strategies are limited to the use of disinfectants and hand sanitizers (which are not highly efficient) or by limiting human exposure through closing down facilities (e.g., hospital wards). A specific and efficacious intervention strategy to treat diarrheal disease and to contain it, preventing virus spreading, is highly desirable. 

Highly potent and effective antivirals have been developed in the last decades against herpes viruses, HIV, HCV and HBV [4]. Specific treatment for acute viral infections such as respiratory infections with RSV are now on the horizon. Achieving effective antiviral treatment for acute diarrhea should thus not be doomed impossible, but needs to be conceptually well-designed, highly efficient and made available timely to the patient. Challenges certainly include the variety of agents but also the installed culture of undervaluing the discomfort and morbidity inflicted by an episode of gastroenteritis. Moreover, patients with primary immune deficiencies, transplant recipients and other immunocompromised individuals can face months of diarrheal episodes thus making this a pressing matter [5,6,7].

It is unrealistic to consider that one antiviral could be developed against each virus that causes AGE, therefore the development of a broad-spectrum antiviral effective against multiple viruses would be preferable and the clinical use would then be made simpler. Rapid laboratory diagnostics should be paired with the start of treatment to ensure the cause is viral and thus the agent susceptible to the treatment. Some broad-spectrum antivirals such as ribavirin are available, but the existence of a syndrome-based single drug (or, eventually, a combination of drugs) would be innovative and highly beneficial to the patient that could finally have access to fast-acting specific treatment for diarrheal disease. Initiating treatment early, i.e., less than two days from the onset of symptoms as is recommended for influenza virus infections [8], would greatly increase the chances of success. On the other hand, for those chronically infected this might be much less critical; a reduction of shedding and symptoms could already be considered a positive outcome. Antiviral prophylaxis would be desirable for those in high risk for severe and prolonged disease, as mentioned above. 

An antiviral drug could be designed with multiple targets in sight, namely the viral surface proteins thus preventing the start of infection or virion release, the viral genome replication machinery hence targeting enzymes such as the polymerase, or cellular factors thus inhibiting virus-host interactions. In this review, we explore the viral polymerase as a target for a syndrome-based approach to antiviral therapy. Structural and functional information on the polymerases of calici-, rota-, adeno- and astroviruses is compared, highlighting common features which can serve as a starting point for the in silico design of novel small-molecule inhibitors with pan-antiviral activity. Potential challenges and limitations to this approach are also discussed.

## 2. The Viral Polymerase as an Antiviral Target

As major diarrhea-causing viruses belong to different viral families, their viral polymerases show substantial differences in functionalities and interactions with other factors to exert their function. Nonetheless, the viral polymerase represents the most conserved target among diverse viral families, as it maintains the same overall structure and functional motifs, despite significant amino-acid sequence variations [9,10]. The viral polymerase is an excellent antiviral target, as proven by the successful antivirals developed against herpesviruses (e.g., acyclovir), HCV (e.g., sofosbuvir), HIV (various nucleoside and non-nucleoside reverse transcriptase inhibitors). Also for RSV (Lumicitabine or ALS-8176) and Ebola virus (Remdesivir or GS-5734) potent nucleoside polymerase inhibitors are in development [11]. This protein therefore represents the most suitable choice to identify broad-spectrum antiviral agents that can block the replication of all, or most, diarrhea-inducing viruses at the same time.

### 2.1. Functional and Structural Comparison of the Viral Polymerase in Diarrhea-Causing Viruses

The viral RNA-dependent RNA-polymerase (RdRp) of noro- and sapoviruses [12], which belong to two genera within the *Caliciviridae*, replicates their (+)ssRNA genome by initiating RNA synthesis according to two different modes, de novo and primed by VPg (viral protein genome-linked) [13]. Sapovirus and norovirus RdRps share high structural analogies, with a sequence similarity of 34% [14]. A crystal structure is available for HSaV in the apo form [14], while several crystal structures are available for human and murine norovirus (MNV) RdRp, including the apo form and multiple complexes with ion cofactors, RNA elements, nucleotide analogues and non-nucleoside inhibitors [9,15,16,17,18,19,20]. The structures of HNoV and HSaV RdRps and their superimposition are illustrated in Figure 1a‒c.

HNoV and HSaV RdRps share a three dimensional organization that resembles a right hand with palm, thumb and fingers domains, and show the presence of six broadly known polymerase-conserved sequence and structural motifs A-F [21,22], along with the motif G and motif H [23]. The RNA template, primer and incoming NTP bind to the large central active site. The two ion cofactors are coordinated in the catalytic site by aspartic acids of the palm subdomain, where the conserved catalytic GDD sequence (motif C) is located. Three allosteric sites have been identified for the HNoV RdRp, according to the binding of co-crystallized non-nucleoside inhibitors [24]. 

HRVs are characterized by a segmented dsRNA and their RNA synthesis takes place within subviral particles called viroplasms. Viral genome replication is carried out, following a particle-associated mechanism, by the viral dsRdRp VP1, which interacts in a replication complex with different nucleic acid and protein elements [25,26,27]. Despite significant functional differences with ssRNA polymerases, VP1 presents the conserved motifs A–H and a right-hand shape, as confirmed by different crystal structures available for the apoenzyme and complexes with RNA and GTP (Figure 2) [27].

Similarly to *Caliciviridae*, HAstVs are characterized by a (+) ssRNA genome, but they represent one of the least studied enteric viruses, and much has to be determined for their viral proteins. Their RdRp is encoded in the open reading frame (ORF) 1b, and it belongs to the supergroup 1 of RNA-polymerases [28], relying on the VPg to initiate transcription [29,30]. Crystal structures are available only for HAstV capsid elements [31,32,33,34]. In order to compare the structure of HAstV RdRp to the other viral polymerases, we have used the RdRp domain from the HAstV-1 non-structural polyprotein sequence Q67726 [35] to run a DELTA-BLAST search [36] within the Protein Data Bank [37], to find the most homologous proteins with an associated crystal structure. Among the closest viral RdRps found were the ones of Sapporo virus (HSaV GI) (16% sequence identity) and Norwalk virus (HNoV GI) (18% sequence identity), which we chose as the template (PDB ID 3BSO [15]) to prepare a homology model for astrovirus RdRp using MOE2018.10 [38]. The sequence alignment, manually adjusted from the one obtained on Clustal Omega [39], and the structure of the final model are shown in Figure 3.

The Ramachandran plot (data not shown) confirmed that all residues of the catalytic site in our model are in the allowed region for their backbone dihedral angles, indicating its reliability to assess in silico potential nucleoside inhibitors targeting the active site. Such models cannot be used with confidence to evaluate allosteric sites.

Differently from the other diarrhea-causing viruses, HAdVs have a linear dsDNA genome, replicated by the viral DNA polymerase (HAdV Pol) according to a protein-primed mechanism [40]. HAdV Pol belongs to the B-family of DNA polymerases, and it contains an exonuclease domain responsible for its proofreading function. Differently from RNA polymerases, HAdV Pol also presents two extra subdomains, TPR1 and TPR2, whose function is not completely understood. HAdV Pol forms a replication complex with viral and cellular factors to produce viral genome duplexes [41], with a mechanism similar to the DNA replication of bacteriophages, including the Bacillus subtilis bacteriophage f29 [42]. While a crystal structure is not available for HAdV Pol, structural data is available for the bacteriophage f29 DNA polymerase [43], and we used its ternary complex with a DNA primer and substrate (PDB 2PYL) to prepare a homology model for HAdV Pol from the sequence of HAdV serotype 40 (UniProtKB entry P48311), using MOE2018.10 [38]. The two protein sequences were manually aligned to match reported functional alignments (Figure 4a) [44], and the final model for HAdV Pol is shown in Figure 4b. As with our HAstV RdRp model, also this one can be used to evaluate the binding of nucleoside inhibitors, but not to evaluate allosteric ones. 

While it remains challenging to identify a broad-spectrum agent that targets all five polymerases, some overall structural similarities might be exploited to attempt to design such a pan-polymerase inhibitor. Although identifying an allosteric site common to all polymerases appears unfeasible, the enzyme active site, in particular the incoming nucleotide subsite, is relatively conserved across the different viruses (Figure 5). This represents a promising starting point to design broad-spectrum nucleoside inhibitors.

### 2.2. Known Polymerase Inhibitors of Diarrhea-Causing Viruses

Different polymerase inhibitors have been found to be active against human noro-, sapo-, rota- and adenoviruses. Until now, no replication inhibitors have been reported for HAstV. The only explored antiviral agent, ribavirin, shows no clear efficacy in case reports [45,46].

#### 2.2.1. Nucleoside Inhibitors

Nucleoside analogues are generally phosphorylated by cellular enzymes to their triphosphate active form, which is incorporated by the viral polymerase in the growing nucleic acid strand, leading to chain termination or viral mutagenesis [47,48]. The nucleoside CMX521 (Figure 6) is the first drug candidate to be moved to phase I trials for the treatment of norovirus infections. It is active against multiple norovirus genotypes and effective in MNV-infected mice treated orally [49]. Ribavirin (RBV) has shown some inhibitory effect against NoV in vitro [50], but failed to demonstrate a clear benefit to patients [7]. Favipiravir (T-705), marketed in Japan to treat influenza [51], and its analogue T-1105 inhibit MNV replication in cells [52,53], and showed variable efficacy in mouse models [54,55]. Other nucleoside analogues active against MNV and HNoV replication are 2’-C-methyl-cytidine (2CMC) [56,57], and its fluorinated analogue 2’-fluoro-2’-C-methyl-cytidine (2’-F-2CMC), the parent nucleoside of mericitabine [58]. Moreover, 2CMC and the 7-deaza-2’-C-methylated analogue of adenosine (7DMA) also inhibit the replication of sapovirus and rotavirus [59], hinting that the proposed broad-spectrum inhibition of multiple viral polymerases is feasible. Finally, 5-nitro cytidine triphosphate (NCT), whose parent nucleoside inhibits poliovirus and coxsackievirus B3 [60], and 2’-amino cytidine triphosphate (ACT) have been co-crystallized with the HNoV RdRp [15,16].

Few nucleoside analogues have been investigated also for their anti-rotavirus activity. 7DMA and 2CMC inhibit in vitro HRV replication with a direct effect on the RdRp, as happens with HNoV [59]. Ribavirin, 3-deazaguanine, 3-deazauridine and 9-(S)-(2,3-dihydroxy-propyl)adenine show an effect which does not appear to be related with the viral RdRp [61], while the old antiviral foscarnet (PFA), a pyrophosphate-mimic [62], adenosine-9-β-D-arabinofuranoside 5’-triphosphate (Ara-ATP), a replication inhibitor for both RNA and DNA viruses, cytosine-9-β-D-arabinofuranoside 5’-triphosphate (Ara-CTP) and 3’-deoxyadenosine 5’-triphosphate [63] inhibit rotavirus RNA synthesis.

Even though no treatment is approved for HAdV infections [64], different nucleos(t)ide analogues inhibit adenovirus replication: cidofovir ((S)-HPMPC, Figure 7) [65], its lipid-ester pro-drug brincidofovir [66], (S)-HPMPA, 2’-nor-cyclic-GMP [67], (S)-HPMPO-DAPy, 2-amino-7-(1,3-dihydroxy-2-propoxymethyl)purine (1), zalcitabine (ddC), alovudine (FddT) [68], stampidine and compound 2 in Figure 7 [69], ganciclovir [67], and finally the ribosylated 6-azacytidine [70].

#### 2.2.2. Key Features of an Ideal Broad-Spectrum Nucleoside Inhibitor

An ideal broad-spectrum nucleoside inhibitor of the described viruses should target both DNA and RNA polymerases. Structurally, the sugar moiety should be flexible enough to adapt to the NTP pocket of both enzyme classes, limiting the choice to few options, mainly acyclic. This option would be supported by the broad-spectrum activity against different RNA viruses found for acyclic fleximer nucleosides [71]. Alternatively, β-D-arabinofuranose could be considered, as suggested in Figure 8. The nucleobase should be chosen from those showing the broadest antiviral activity, such as the fluorinated pyrazine-carboxamide of T-705 (RNA viruses), or a modified cytidine (DNA viruses). To enhance their cell-permeability and conversion to the active tri-phosphate forms, such compounds should be prepared in the form of pro-drugs, such as the McGuigan Pro-Tide (Figure 8) [72], present in the blockbuster anti-HCV drug sofosbuvir [73].

The triphosphate forms of the suggested nucleoside analogues have been analyzed with docking simulations [74] on the NTP binding pocket of the HNoV RdRp, as representative for RNA viruses, and on the active site of our model for HAdV Pol. As exemplified for one potential structure (Figure 9), docking results confirm the potential of these analogues to fit the active pocket of both polymerases.

#### 2.2.3. Non-Nucleoside Inhibitors

To date, non-nucleoside inhibitors targeting the viral polymerase have been reported only for norovirus (Figure 10), and some have been co-crystallized with the enzyme, allowing the identification of three allosteric sites in NoV RdRp. Suramine and NF023 bind the NTP access pathway between fingers and thumb sub-domains [19], naphthalene di-sulfonate NAF2 binds to a sub-pocket of the thumb domain [24], while PPNDS binds to a second sub-region of the thumb domain [20]. These compounds have no activity against the viral replication in cells, possibly due to poor cell permeability [19], while the HNoV RdRp inhibitor NCI02 inhibits both HNoV and MNV replication [75], and NIC12, a weak inhibitor of poliovirus polymerase [76], weakly inhibits MNV replication in cells [75]. Different allosteric inhibitors of HCV polymerase were recently evaluated as inhibitors of *Caliciviridae* RdRps [77]: JTK-109, TMC-647055 and Beclabuvir (Figure 10) inhibited six *Caliciviridae* RdRps, spanning *Norovirus*, *Sapovirus* and *Lagovirus*. JTK-109 also inhibited MNV replication in cell-based assays, providing a potential starting point to develop pan-RdRp inhibitors.

## 3. Challenges and Potential Limitations to This Approach

### 3.1. In Vitro and In Vivo Replication Systems Available for Diarrhea-Causing Viruses

One of the main reasons for the lack of antiviral therapies against viral AGE, is the lack of suitable cell culture systems and/or animal models for the majority of these viruses. The HNoV is not easily cultivated in vitro or in vivo, therefore most antiviral research is being performed on the MNV or the HNoV GI replicon. Only recently it was reported that HNoV can replicate in the human B-cell line BJAB and in stem-cell-derived enteroids [78,79]. These models were a first breakthrough in cultivating the HNoV but further optimization would facilitate their use in drug discovery campaigns. For HSaV there is no in vitro or in vivo replication system available. The porcine SaV Cowden strain can infect gnotobiotic pigs and porcine kidney cells in the presence of bile acids [80,81]. Multiple strains of rotaviruses can be cultivated in vitro in the presence of trypsin; in vivo models to study rotavirus infections are rather limited [82,83,84]. HAdVs type 40 and 41 have limited ability to replicate in cells, when compared to other adenovirus subtypes, plus animal models are lacking [85]. Most HAstV genotypes grow in cell culture [86] but there is no small animal model available. A murine astrovirus model in immunodeficient mice has been reported [87], but the most widely used in vivo model are turkey poults, which are infected with the turkey astrovirus [88].

One advantage of developing polymerase-targeting inhibitors is the availability of enzymatic assays which allow the initial optimization of small molecule inhibitors, which can go into cellular assays at a later stage. These are available for multiple norovirus genotypes, for sapovirus and adenoviruses, but not for astroviruses [19,89]. In the case of rotavirus, polymerase activity can be assessed using purified viroplasms containing the active polymerase-capping enzyme complex VP1-VP3 [59]. Nonetheless, the limited availability of models is a limitation for drug discovery efforts, also because these would help to further understand the viral life cycles thus providing important insights for the development of antiviral therapy.

### 3.2 Antiviral Drug Resistance

Viral replication usually has a high error rate, causing the generation of resistant mutants able to evade a given treatment, in particular if administered long-term. This has been observed with early anti-HIV reverse transcriptase inhibitors, but later overcome with combination therapies of drugs belonging to different classes, with high genetic barrier to resistance, i.e., requiring multiple mutations for the virus to become resistant [90]. Also for HCV, multi-drug treatment regimens (most of which now include sofosbuvir) are characterized by a high barrier to resistance [91], allowing the suppression of most HCV genotypes [92].

Although all direct-acting antiviral agents can lead to resistance, this issue would have a different impact in the case of the acute infections, as gastroenteritis mostly is. Treatment of acute infections aims to reduce virus replication enough to allow the immune system to clear the virus, and is therefore of short duration [93]. The incidence and clinical impact of resistance development would therefore be different and less likely to occur than for chronic infections.

In addition, resistant viral strains are usually less virulent than the wild type [94], and acute viral infections still resolve in a self-limited manner. Even when treating prolonged/chronic norovirus infections, the treatment course(s) would be expected to be short, potentially rendering the issue of resistance less problematic in this context.

### 3.3. Mitochondrial Toxicity

Several drugs cause mitochondrial toxicity as a side effect, potentially leading to tissue damage and organ failure. The main risk of mitochondrial toxicity for antivirals acting on the viral polymerase or reverse-transcriptase is due to inhibition of the mitochondrial γDNA-polymerase [95] or RNA-polymerase POLRMT [96]. This side-effect on the γDNA-polymerase is relevant for the anti-HIV nucleoside reverse-transcriptase inhibitors zalzitabine (ddC), didanosine (ddI), stavudine (d4T), zidovudine (AZT), lamivudine (3TC), abacavir (ABC) and tenofovir (TFD) [97,98]. Similarly, different antiviral ribonucleoside analogs show toxic effects due to their affinity for POLRMT. In particular, valopicitabine, an ester prodrug of 2CMC in phase 2 clinical trials for HCV, was not advanced further due to gastrointestinal toxicity [99], while the triphosphate active forms of 2’-C-methyl nucleosides 2CMC, 2CMA and 2CMG inhibit POLRMT by chain termination [100], indicating a potential underlying issue for the development of any 2‘-C-methylated nucleoside into a drug. However, the triphosphate form of 2CMU (uridine) does not impact POLRMT function [96], while sofosbuvir, which combines a 2’-C-methyl feature with a 2’-fluorine substitution on the sugar, does not cause any mitochondrial inhibition [101]. General evidence appears to indicate that 2’-monosusbtitutions, along with 4’-monosubstitutions, on the sugar of antiviral ribonucleosides do not provide selectivity for the viral polymerase over POLRMT, with the associated mitochondrial toxicity mainly due to the core nucleoside rather than the cleaved prodrug moieties [97]. Nonetheless, structure-activity relationships for mitochondrial toxicity remain unpredictable, and this off-target effect can potentially be avoided with tuned modifications at both the sugar and the nucleobase level of selected nucleoside analogues. Potential mitochondrial toxicity should thus be carefully evaluated for each single nucleoside analogue of interest, possibly following a recently developed screening method which combines together multiple biochemical and cellular assays to assess this undesired effect [97].

## 4. Concluding Remarks

Almost one hundred antivirals have been marketed since 1963, however mostly to treat one specific human viral disease [4]. Given the diversity of human viral pathogens, there is the need to focus on broad-spectrum inhibitors against viral families or even all RNA or DNA viruses. We here explored this approach for rota-, noro-, sapo-, astro-, and adenovirus. In this review, we have emphasized the viral polymerase by studying the functional and structural similarities between the selected polymerases, as in our perspective this offers a starting point for broad-spectrum therapy against diarrhea-causing viruses. The next step would be an in silico structure-based screening of nucleoside and non-nucleoside small-molecules, using a series of molecular docking simulations on the crystal structures or homology models of the five polymerase discussed. This strategy would allow the identification of chemical scaffolds with good predicted affinity for all proteins at the same time, while their subsequent assessment in enzymatic inhibition assays would provide potential broad-spectrum hits for further optimization.

## Figures and Tables

**Figure 1 viruses-11-00173-f001:**
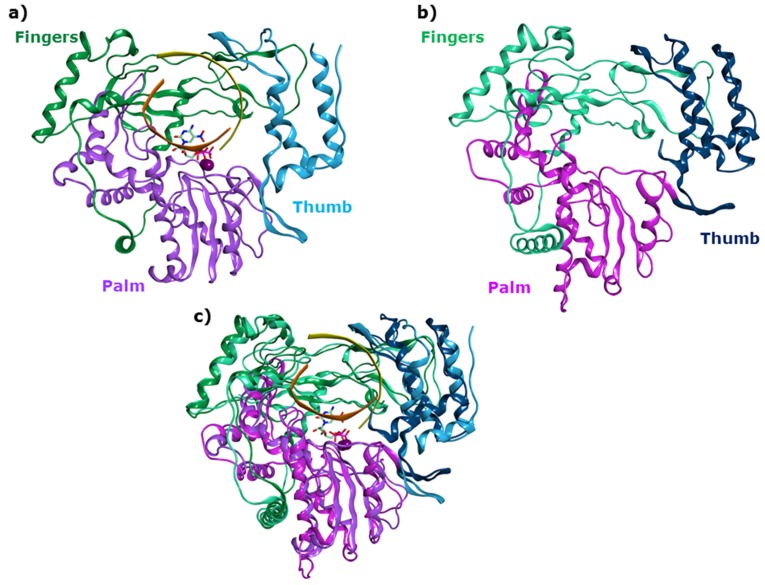
(**a**) Three-dimensional structure and domains of HNoV RdRp (Norwalk virus, GI.1) in complex with RNA template strand, RNA growing strand and 5-nitrocytidine (PDB ID 3BSN, resolution 1.8 Å); (**b**) HSaV RdRp (Sapporo virus, Hu/SV/Man/1993/UK) in its apo form (PDB ID 2CKW, resolution 2.3 Å); (**c**) structural superimposition of the two RdRps.

**Figure 2 viruses-11-00173-f002:**
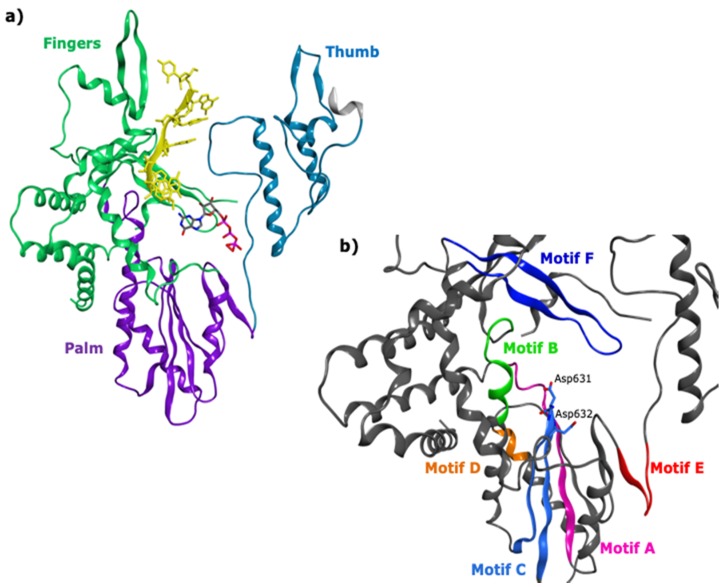
(**a**) Domains of the HRV VP1 co-crystallised with an RNA fragment and GTP (PDB ID 2R7X, resolution 2.8 Å); (**b**) colour-coded motifs A-F in the 2R7X crystal structure, with the catalytic residues Asp631 and Asp632 highlighted (part of the conserved GDD sequence).

**Figure 3 viruses-11-00173-f003:**
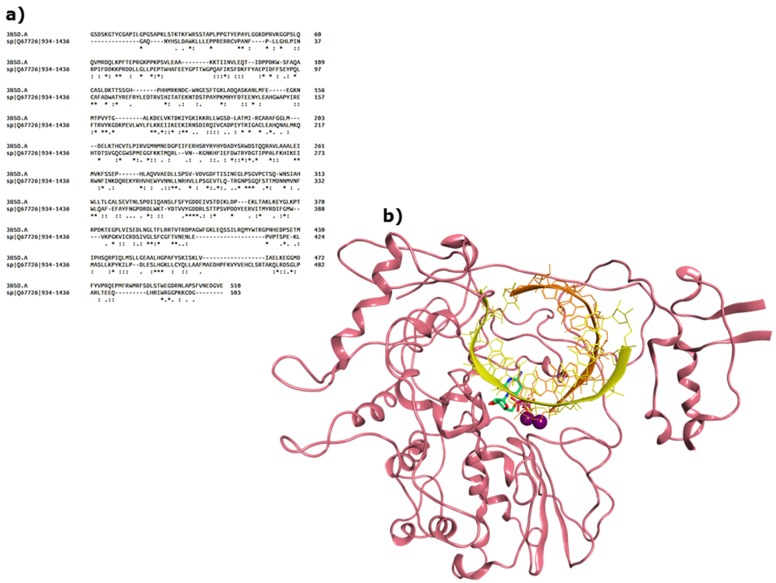
(**a**) Sequence alignment between HNoV RdRp from the 3BSO crystal structure and HAstV RdRp portion in the Q67726 sequence; (**b**) final model for HAstV RdRp.

**Figure 4 viruses-11-00173-f004:**
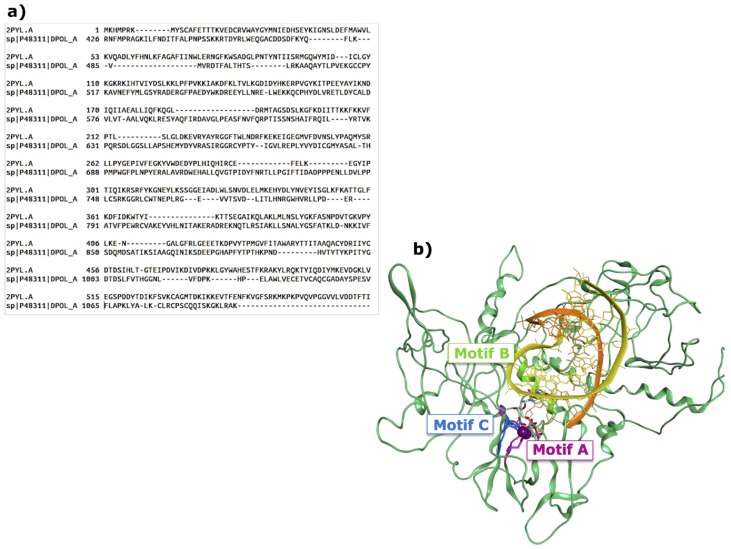
(**a**) Sequence alignment between the bacteriophage f29 DNA polymerase and the polymerase domain of HAdV F serotype 40; (**b**) final model for the AdV Pol, including the nucleic acid and NTP components minimized from the template.

**Figure 5 viruses-11-00173-f005:**
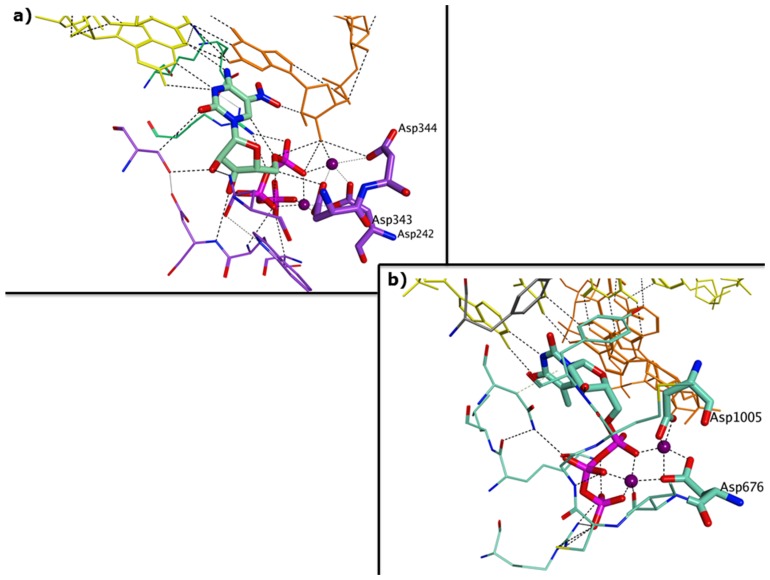
Similarities between RNA and DNA polymerases of diarrhea-causing viruses. (**a**) Incoming nucleotide binding pocket of the ternary complex of HNoV RdRp co-crystallised with 5-nitro cytidine triphosphate (PDB ID 3BSN); (**b**) incoming nucleotide binding pocket of the polymerase homology model obtained for HAdV.

**Figure 6 viruses-11-00173-f006:**
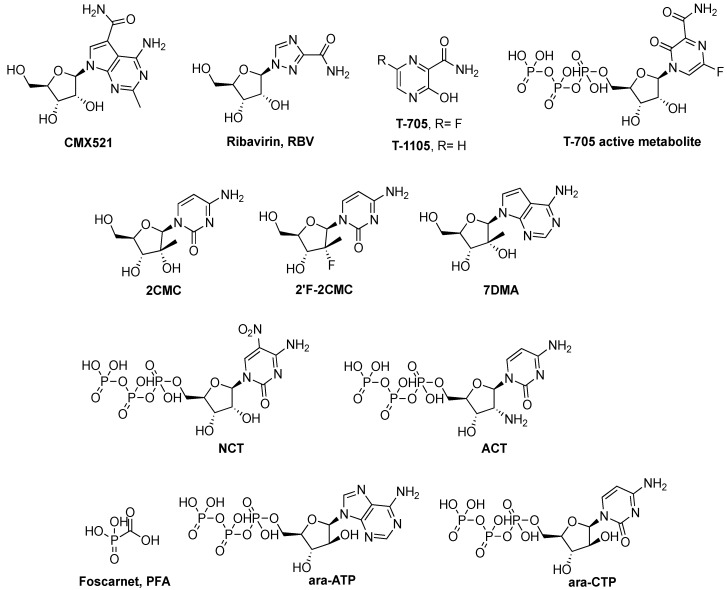
Structure of reported nucleoside analogues (or pyrophosphate mimics) inhibiting the RdRps of caliciviruses or rotaviruses.

**Figure 7 viruses-11-00173-f007:**
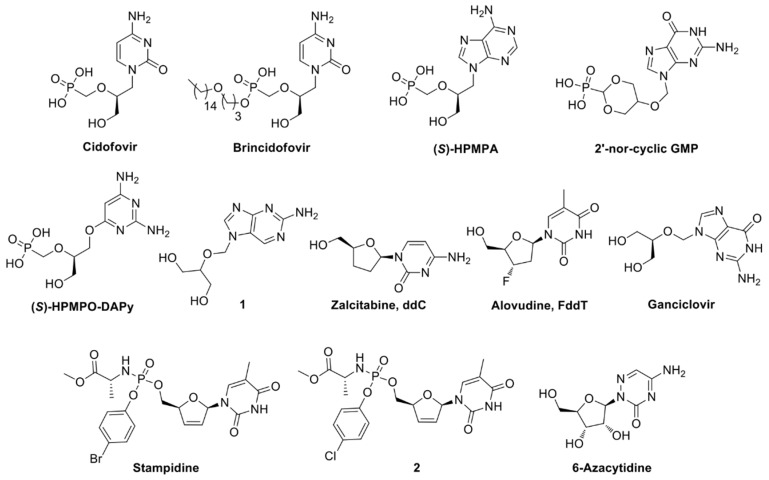
Nucleoside analogues inhibiting HAdV pol/HAdV replication.

**Figure 8 viruses-11-00173-f008:**
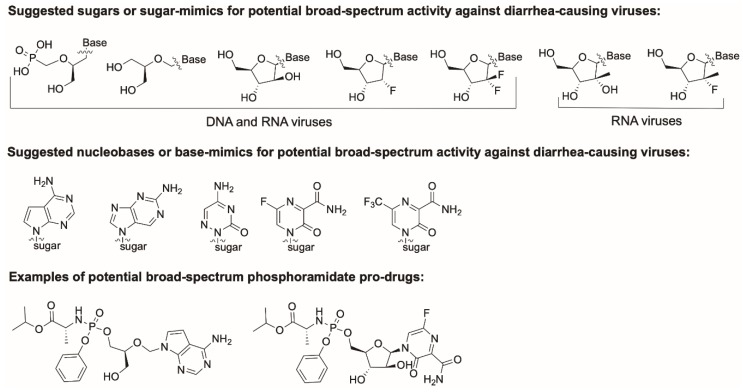
Potential sugar and nucleobase modifications for broad-spectrum nucleoside inhibitors.

**Figure 9 viruses-11-00173-f009:**
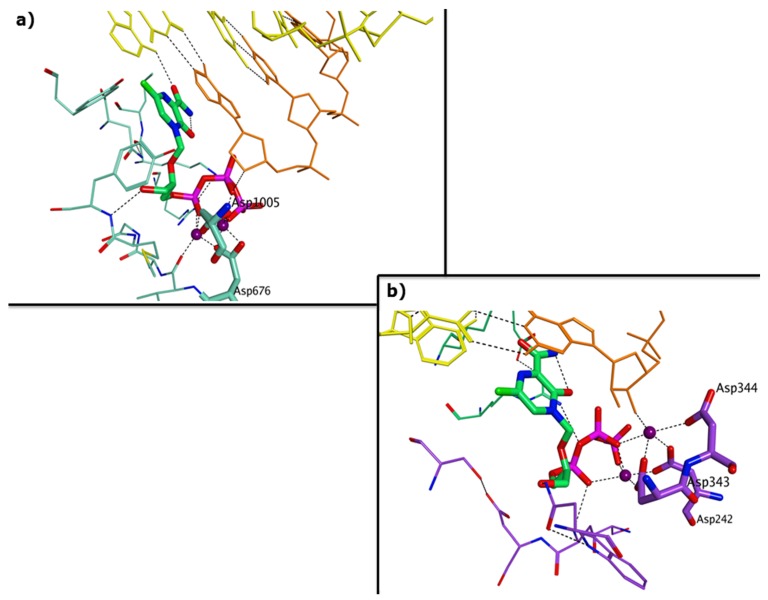
Predicted binding of one suggested potential broad-spectrum polymerase inhibitor to the HAdV Pol model (**a**) and to HNoV NTP binding pocket in the 3BSN crystal structure (**b**).

**Figure 10 viruses-11-00173-f010:**
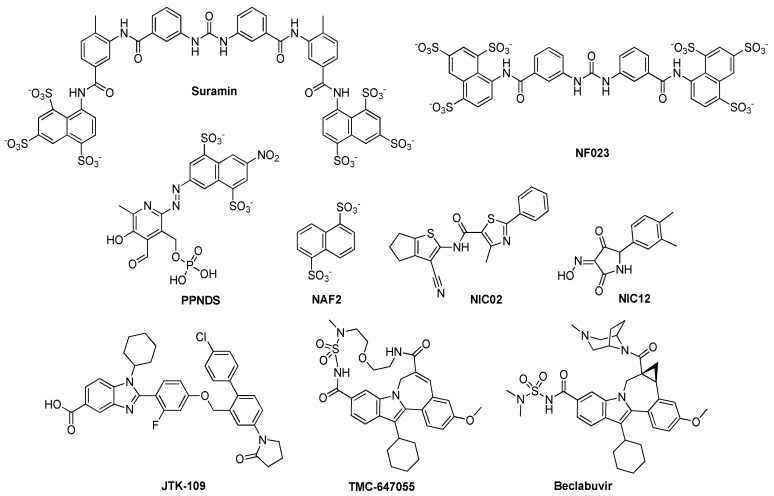
Structures of reported non-nucleoside inhibitors of the *Caliciviridae* RdRp.

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
