# Peer review of "Targeting the Viral Polymerase of Diarrhea-Causing Viruses as a Strategy to Develop a Single Broad-Spectrum Antiviral Therapy"

_viruses, 2019, doi:10.3390/v11020173_

Reviewer 1 Report

This is a terrific review from Rocha-Pereira and colleagues on the identification of a single broad antiviral therapy to control all diarrhoea-causing viruses (e.g. norovirus, rotavirus, adenovirus, astrovirus, adenovirus). The manuscript is quite a comprehensive work as it entails detailed structural and functional information on different viral polymerases (including RNA and DNA templated replicases), drugs showing activity against each different pathogen, cell culture and animal models available (and limitations associated), major toxicity issues (mitochondrial polymerases) and strategies to overcome them.

I also consider that the text is very well written and exceptionally well organised, making very enjoyable its reading. I find the topic will be of interest to may researchers on the field of antiviral research, viral replication and structure-function of polymerases. Thus, I strongly recommend its rapid communication in Viruses. I only have very minor suggestions for the authors to consider:

 In addition to broadly known A-F motifs they have been identified two additional conserved regions (G and H) in RdRps. There are several papers on this topic, e.g. Venkataraman et al 2018. Viruses. 10, pii E76

In Concluding remarks I have missed a couple of sentences where the authors may want to discuss what possible strategies should be pursued and to achieve the overall aim stated in the title: developing a single broad antiviral therapy against viral diarrhoea diseases. What antiviral drugs or combination of them are most likely to lead to a pan-antiviral activity against diarrhoea viruses?

Line 254. less virulent than

Lines 47-50 efficacious is repeated three times in a short paragraph. Can you use alternative synonyms here?

Author Response

1.       In addition to broadly known A-F motifs they have been identified two additional conserved regions (G and H) in RdRps. There are several papers on this topic, e.g. Venkataraman et al 2018. Viruses. 10, pii E76.

The information on the motifs G and H has been included in the manuscript, together with the suggested reference. (line 102-103, 114)

 2.       In concluding remarks, I have missed a couple of sentences where the authors may want to discuss what possible strategies should be pursued and to achieve the overall aim stated in the title: developing a single broad antiviral therapy against viral diarrhea diseases. What antiviral drugs or combination of them are most likely to lead to a pan-antiviral activity against diarrhea viruses?

A comment on how to develop such broad-spectrum agent has been included in the concluding remarks. (lines 290-298)

 3.       Line 254. less virulent than

This has been corrected in the manuscript. (line 254)

 4.       Lines 47-50 efficacious is repeated three times in a short paragraph. Can you use alternative synonyms here?

Alternative synonyms are now used in this paragraph of the manuscript. (lines 47-50)

Reviewer 2 Report

This review deals with a strategy to develop a small-molecule inhibitors with pan-antiviral activity against gastroenteritis-related viruses (Rotavirus, Norovirus, Sapovirus, Astrovirus, and Adenovirus). The authors describes the viral polymerase as a target of antiviral therapy. They present that structural and functional information on the polymerases of gastroenteritis-related viruses may serve as the in silico design of novel broad-spectrum antiviral therapy. This review is well written and illustrated. I have annotated the manuscript with several minor corrections, which I believe will improve the readability of the paper.

 Figures 1-4. Please show the accession numbers of amino acid sequence cited by the authors to predict three-dimensional structure of viral polymerases.

 Figure 1. Please specify the genogroups and genotypes.

 Figures 1 and 2. The same analysis should be done for astrovirus and adenovirus.

 Figures 3 and 4. The same analysis should be done for norovirus and rotavirus.

 Lines 158-215. The authors should summarize the antiviral drugs of gastroenteritis-related viruses in tables.

Author Response

1.       Figures 1-4. Please show the accession numbers of amino acid sequence cited by the authors to predict three-dimensional structure of viral polymerases.

The authors believe that this information is already present, as all figure legends explicitly report the sequence ID of the amino acid sequences, or PDB ID of the structures used, as specified below:

Figure 1&2: crystal structures are shown, not models, and their PDB ID is specified in figure legend.

Figure 3&4: sequence ID’s are specified both in legend and actual alignment (Figure 3a).

 2.       Figure 1. Please specify the genogroups and genotypes.

The information has been added in the legend of Figure 1.

 3.       Figures 1 and 2. The same analysis should be done for astrovirus and adenovirus. Figures 3 and 4. The same analysis should be done for norovirus and rotavirus.

In our opinion, repeating the same analysis as suggested would be redundant, considering the information given and the pictures shown, and would not add valuable information to the manuscript.

 4.       Lines 158-215. The authors should summarize the antiviral drugs of gastroenteritis-related viruses in tables.

In our opinion, adding such table is not within the scope of this review, since the focus in on the antiviral target, not on described inhibitors. Besides, extensive reviews on antiviral drugs for gastroenteritis are already available. For example, this very recently published review describing all potential therapeutics that have been reported for norovirus infections by NE. Netzler et. al. (Med Res Rev, 2018 Dec 25, doi: 10.1002/med.21545).

Reviewer 3 Report

Targeting the viral polymerase of diarrhea-causing viruses as a strategy to develop a single broad-spectrum antiviral therapy

 This review summaries the functional and structural similarities of viral polymerases from viruses that are responsible for gastroenteritis (human rotavirus, norovirus, astrovirus, sapovirus and adenoviruses). This manuscript is well-written and clearly presented and the information in the manuscript with an emphasis on enteric viruses seems to be a good addition to the growing body of research on finding broad-spectrum antiviral compounds for multiple viruses. The reviewer suggests only minor changes in the manuscript.

Minor revision:

Line 63: ‘some broad-spectrum antivirals such as ribavirin are available’ - please consider adding a short description on ribavirin’s inadequate potency for gastroenteritis viruses (this appears later in the manuscript in line 157)

 Line 158-216: Please consider including some information on the potency of each antiviral compounds appeared in the text (IC50 or EC50).

 The authors included in the manuscript the available crystal structures (human norovirus, sapovirus and rotavirus) or structural modelling of viral polymerase (astrovirus and adenovirus). The authors also included Fig. 5 to show the similarities between the binding pockets of human norovirus and human adenovirus polymerases. The reviewer suggests adding a panel of overlapping structures of these five viruses in the catalytic site in figure 5.

Author Response

1.       Line 63: ‘some broad-spectrum antivirals such as ribavirin are available’ - please consider adding a short description on ribavirin’s inadequate potency for gastroenteritis viruses (this appears later in the manuscript in line 157).

As the reviewer mentions, this information is provided later in the manuscript. In our opinion it is sufficient to mention its potency against diarrhea-causing viruses only once in this review.

 2.       Line 158-216: Please consider including some information on the potency of each antiviral compounds appeared in the text (IC50 or EC50).

In our point of view adding this information goes beyond the scope of the manuscript, which is to focus on the target structures and at the chemical structures of potential broad-spectrum inhibitors, not on the specific potency of known inhibitors against single viruses. Furthermore, the appropriate references were added, so those readers who are interested in that information can easily find it by consulting the original manuscript.

 3.       The authors included in the manuscript the available crystal structures (human norovirus, sapovirus and rotavirus) or structural modelling of viral polymerase (astrovirus and adenovirus). The authors also included Fig. 5 to show the similarities between the binding pockets of human norovirus and human adenovirus polymerases. The reviewer suggests adding a panel of overlapping structures of these five viruses in the catalytic site in figure 5.

Even though this suggestion is very appropriate, unfortunately the structural alignment of all five proteins at the same time in a single picture is not feasible, mainly due to the presence of a DNA-polymerase in the requested alignment. The resulting figure would not add any valuable information, as it would result very crowded and potentially misleading, since it would not be straightforward to identify the similarities highlighted instead in Figure 5 in its present state. We therefore did not include such figure in the revised manuscript.